# Promoter-Bound Full-Length Intronic Circular RNAs-RNA Polymerase II Complexes Regulate Gene Expression in the Human Parasite *Entamoeba histolytica*

**DOI:** 10.3390/ncrna8010012

**Published:** 2022-01-27

**Authors:** Jesús Alberto García-Lerena, Gretter González-Blanco, Odila Saucedo-Cárdenas, Jesús Valdés

**Affiliations:** 1Departamento de Bioquímica, CINVESTAV-México, Av. IPN 2508 Colonia San Pedro Zacatenco, Mexico City 07360, Mexico; alberto.lerena@cinvestav.mx (J.A.G.-L.); gretter.gonzalez@cinvestav.mx (G.G.-B.); 2Departamento de Histología, Facultad de Medicina, Universidad Autónoma de Nuevo León, Monterrey 67700, Mexico; odilam@gmail.com

**Keywords:** flicRNA, circRNA, Pol II-CTD, transcription regulation, gene promoter, splicing

## Abstract

Ubiquitous eukaryotic non-coding circular RNAs are involved in numerous co- and post-transcriptional regulatory mechanisms. Recently, we reported full-length intronic circular RNAs (flicRNAs) in *Entamoeba histolytica*, with 3′ss–5′ss ligation points and 5′ss GU-rich elements essential for their biogenesis and their suggested role in transcription regulation. Here, we explored how flicRNAs impact gene expression regulation. Using CLIP assays, followed by qRT-PCR, we identified that the RabX13 control flicRNA and virulence-associated flicRNAs were bound to the HA-tagged RNA Pol II C-terminus domain in *E. histolytica* transformants. The U2 snRNA was also present in such complexes, indicating that they belonged to transcription initiation/elongation complexes. Correspondingly, inhibition of the second step of splicing using boric acid reduced flicRNA formation and modified the expression of their parental genes and non-related genes. flicRNAs were also recovered from chromatin immunoprecipitation eluates, indicating that the flicRNA-Pol II complex was formed in the promoter of their cognate genes. Finally, two flicRNAs were found to be cytosolic, whose functions remain to be uncovered. Here, we provide novel evidence of the role of flicRNAs in gene expression regulation in cis, apparently in a widespread fashion, as an element bound to the RNA polymerase II transcription initiation complex, in *E. histolytica*.

## 1. Introduction

CircRNAs are a type of widely conserved transcripts in eukaryotic cells, first identified by electron microscopy and sedimentation studies in plant viroids [1] and as exonic circRNAs in the cytoplasm of HeLa cells [2]. These molecules are highly stable due to their resistance to exonucleases, and they are capable of interacting with proteins and other RNAs [3]. CircRNAs can be monoexonic, multi-exonic, exon-intronic, or intronic. Exonic circRNA biogenesis occurs by co-transcriptional back splicing, involving a downstream 5′ss donor covalently linked to an upstream 3′ss acceptor, thus producing a circular molecule. Back splicing may also occur post-transcriptionally from a skipped exon within a lariat made from linear splicing resulting in exon-intronic circRNAs (EIciRNAs) [4,5,6].

Additional spliceosomal and cis-elements could mediate back splicing. Alu repeats in flanking introns to an exon or a group of exons allows the formation of a loop that facilitates circularization [7,8]. CRISPR/Cas9 deletion of such Alu repeats prevented circularization [9]. Protein–protein interaction constitutes an alternative mechanism of circularization. RNA binding proteins (RBP) QKI, the splicing factor MBL, and FUS proteins have been associated with the circRNAs biogenesis [10,11,12]. Furthermore, intronic circRNAs are covalently closed lariat products resulting from intron lariat processing [6,13].

Depending on their localization, circRNAs have been associated with different functions. Those with exonic sequences have been localized mainly in the cytoplasm and associated with regulatory processes such as microRNAs and protein sponging, regulation of translation, mRNA stability, and 5′ cap-independent translation [10,14,15,16,17,18]. Intronic circRNAs show more heterogeneous localization inside the cell. In the nucleus, they have been linked to the regulation of gene expression, while the function of cytoplasmatic intron circles remains unknown [19,20]. Nuclear EIciRNAs were reported as part of the RNA polymerase II (Pol II) binding complex to downregulate their parental genes [4,21]. For instance, insights on the function of the human 2′–5′ intronic circRNAs (ciRNAs), ci-ankrd52 and ci-sirt7, showed that these ciRNAs directly associate to the serine 2-phosphorylated C-terminus domain of Pol II (CTD-Pol II), indicating that ciRNA-Pol II-CTD interaction occurs during transcription elongation [21].

Topologically, a lariat becomes a circular intronic molecule after trimming the 3′-end tail with a short lifetime following its release from the spliceosome and before its hydrolysis by the debranching enzyme (Dbr1). Nevertheless, several introns show cis-elements that confer stability to them and accumulate inside the cell, such as those classified as stable sequence introns (sisRNAs) [19], including 2′–5′ covalently linked intronic circular RNAs which are processed lariats, and 3′–5′covalently-linked species with untrimmed 3′-end tails.

Recently, we identified full-length intronic circular RNAs (flicRNAs) derived from several transcripts of the protozoan parasite *Entamoeba histolytica* [22]. In agreement with previous reports [23], the 5′ss and 3′ss ends of these molecules were covalently linked with an average of 10 nucleotides between the branch point (BP) and the 3′ss in their lariat precursors [22]. We observed that flicRNAs were post-splicing products originating from the lariat by mechanisms not fully understood. In vivo splicing assays linked the conserved GU-rich 5′ss with the biogenesis of flicRNAs, and the mutation of this element increased RabX13 gene (EHI_065790) expression due to decreased flicX13 production, suggesting that flicRNAs might regulate the expression of their parental genes [22]. Here, we explored some features by which flicRNAs exert transcriptional regulation. CLIP assays showed the association of different flicRNAs from virulence-associated loci to the Pol II-CTD in the parasite *E. histolytica*. Inhibition of the second step of splicing using boric acid (BA) reduced flicRNA formation, thus affecting the expression of both their parental genes and unrelated genes, in a similar way observed in transfectants with a deficient debranching and RabX13 mutated introns [22]. Double transfectants with RabX13 mutations showed an inhibition of flicRNA-Pol II-CTD association. Finally, we showed that the flicRNA-Pol II complex localizes to the core promoter of their parental genes and genes associated with cell cycle regulation. Our results allow us to suggest that flicRNAs participate in a broad network of gene expression regulation in this protozoan.

## 2. Results

### 2.1. CLIP Assays Reveal flicRNAs-RNA Polymerase II Interactions

CLIP and ChIP assays were used to assess flicRNAs-Pol II association in solution and on the promoters of different genes. To this end, we first established hemagglutinin (HA)- tagged amoeba transformants of the Pol II-CTD (to monitor flicRNA-Pol II-CTD association), as well as the full-length major subunit of Pol II (to observe promoter-bound flicRNA-Pol II associations). Assays were compared to the empty vector (pExEhHA) *Entamoeba* transformant controls or to HA-tagged amoeba transformants with decreased or increased RabX13 flicRNA expression: respectively, ΔGU (RabX13 minigene 5′ss intronic mutant) and Dbr1ΔC (intron lariat debranching-deficient enzyme) [22]. To control efficient CLIP signal detection, the expression of the tagged genes in the single and double-transfectant strains was first verified by RT- PCR using primers to detect HA-Pol II-CTD and HA-ΔGU DNA fusions (Figure 1A).

Using immunofluorescence and confocal microscopy assays, we next showed that the expressed Pol II-CTD conserved its nuclear localization in transfected trophozoites using FIT-C conjugated secondary antibodies. Whereas no signal was detected in the cytoplasm, significant punctate signals were found in the nucleus of 60% of the cells, as was expected from two nuclear localization signals included in the Pol II-CTD domain insert (Figure 1B).

To validate our CLIP methods and further analyze the implication of the flicRNAs in transcription regulation, the model flicRNA, flicX13, was monitored using outward-facing primers in circular RT-PCR [24] after the CLIP assays were conducted in Pol II-CTD/ΔGU trophozoites that exogenously express both Pol II-CTD and the RabX13 minigene bearing a mutation in the 5′ss of the intron that decreases flicRNA biogenesis [22]. As expected, compared to the empty vector and nuclear input, we observed increased recovery of flicX13 in the Pol II-CTD immunoprecipitations, which correspondingly decreased in the Pol II-CTD/ΔGU double transfectants (Figure 1C). These results validated our approach and indicated that flicRNA-mediated transcriptional regulation might be exerted via flicRNA-Pol II-CTD interaction.

CLIP assays were then carried out to detect flicRNAs of the RabX13 gene, the virulence-associated loci EHI_169670 (both introns), EHI_014170, EHI_192510, and Clc2B, and those of the ribosomal proteins RpS14 and RpL12: flicX13, flic670i1, flic670i2, flic170, flic510, flicCl2B, flicS14 and flicL12, respectively. CLIP specificity was confirmed by the lack of amplification of the mRNA of a non-related gene (Hsp70) as well as the exonic circRNA circ670e2 [25] (exonic circRNAs are mainly cytoplasmic [6]) (Figure 1D). Actin was amplified to normalize circ670e2 amplification between fractions.

We found that flicX13, flic670i1, flic670i2, flic170, and flic510 were bound to the Pol II-CTD, suggesting their participation in regulating RNA Pol II in cis (Figure 1D). To gain insights into this possibility, we reasoned that since U2 snRNA interactions with phospho-Serine 5 (P-Ser5) residues in the Pol II-CTD [26,27] are required for cotranscriptional spliceosome assembly on the nascent pre-mRNA, as well as interactions of U6 snRNA with U2 snRNA and the 5′ss to activate the splicing complex B [28,29], a search for U2 and U6 snRNAs as part of the Pol II cotranscriptional complex was required. A strong signal of U2 snRNA, but not of U6 snRNA (Figure 1D), was observed, suggesting that flicRNA-Pol II-CTD interaction occurred during transcription initiation/elongation before complex B activation.

Since no CLIP signal was detected for flicCl2B, flicS14, and flicL12, we identified flicS14 and flicL2 in the total RNA from the trophozoites (Figure 2A); their absence in the Pol II-CTD immunoprecipitants and the nuclear input suggested a cytoplasmatic localization of these ribosomal flicRNAs. Purified cytosolic RNA was used to perform divergent RT-PCR assays to determine their subcellular localization. Both flicRNAs were detected in the cytoplasmatic fraction (Figure 1E). flicS14 appeared in a band at 73 bp while flicL12 showed an unspecific amplification around 100 bp.

### 2.2. flicRNAs Modify Gene Expression

To explore some mechanisms involved in flicRNA-mediated transcription regulation, Pol II-CTD transfected trophozoites were treated with boric acid (BA, H_3_BO_3_) 5 mM for 2 and 5 h, inhibiting the second step of splicing and reducing flicRNA biogenesis. In agreement with the in vivo studies in *E. histolytica* trophozoites [22], we observed a decrease in the production of flicX13, flic170, flic670i1, flicS14, and flicL12 in transfectants under H_3_BO_3_ treatments (Figure 2A). We treated Pol II-CTD transfectants with BA to link flicRNAs with gene expression and monitored flicX13 and mRabX13 expression by RT-qPCR. We found that the BA treatment sharply decreased flicX13 expression and increased RabX13 mRNA expression (Figure 2B). The decrease of flicX13 expression was comparable to the flicRNA biogenesis-deficient ΔGU transfectants. Conversely, the increase of flicX13 expression was similar to that of Dbr1ΔC amoeba transfectants, which elicit flicRNA accumulation [22].

To discard that BA treatments could affect the transcription of genes other than their own, we monitored the expression of constitutive transcripts. The expression of actin and Hsp70 did not change significantly in H_3_BO_3_ treated amoebas compared to untreated cells (Appendix A).

### 2.3. ChIP Identifies Promoter-Associated flicRNA-Pol II Complexes

To corroborate that the flicRNAs-Pol II complex directly binds to the promoter of their parental and other target genes, we performed ChIP assays in full-length Pol II transfectants, screening the promoter regions of the following genes: RabX13; the virulence-associated loci EHI_169670, EHI_014170, EHI_042870, EHI_083590, EHI_192510; the nuclear distribution protein EhNudC locus EHI_023890; the structural maintenance of chromosomes protein EhSMC locus EHI_187190; and the RING finger-containing putative transcription factor locus EHI_098590 (EhRNF). As before, we confirmed the overexpression of Pol II fusion protein in amoeba transfectants by Western blot using anti-HA tag antibodies (Figure 3A).

ChIP showed the interaction of the Pol II complex with the core promoter of the flicRNA-forming genes RabX13, EHI_169670, EHI_014170, and EHI_192510. On average, the flicRNA-Pol II complex was bound 300 nucleotides upstream of the AUG site (+1), except for locus EHI_014170 that has an additional binding site at −913 nt (Figure 3B). Unexpectedly, ChIP also identified the interaction of the Pol II complex on the promoters of the non-flicRNA-forming genes EhNudC, EhSmc, and EhRNF (Figure 3C), suggesting that flicRNAs might regulate gene expression more broadly.

RNA was isolated from the ChIP elution to corroborate this, and divergent RT-PCR was carried out. The presence of flicX13, flic510, flic670i1, flic670i2, and flic170 (Figure 4A) indicated that flicRNAs interact with Pol II (particularly with the Pol II-CTD, as shown above) in transcription initiation/elongation complexes on the promoters of their parental genes. Furthermore, compared to untreated trophozoites, the diminishing expression of flicRNAs treated with boric acid increased the expression of the EhNudC gene 3-fold. It reduced the expression of the EhSMC and EhRNF genes to one-third (Figure 4B). The same analysis was performed on the virulent flicRNA-forming genes. The expression of flicRNAs from loci EHI_192510 and EHI_169679 was reduced down to 10% and that from locus EHI_014170 increased 6-fold (Figure 4B). These results demonstrate that flicRNAs regulate the expression of both their parental genes and of non-flicRNA-forming genes, suggesting that, in principle, flicRNA-Pol II complexes might regulate the expression of a broad spectrum of genes.

## 3. Discussion

Using CLIP assays, we identified flicRNAs from several genes of *E. histolytica* as part of the RNA polymerase II binding complex. The exonic circRNA circ670e2 and the constitutive transcript Hsp70 were absent from immunoprecipitations, supporting the specificity of the complexes (Figure 1D). Furthermore, flicRNAs participated in the downregulation of transcription of their parental genes in cis (Figure 2A) and appeared to impact transcription in a general fashion (Figure 4). Zhang et al. [21] obtained similar results studying the association of ciRNAs with Pol II by RIP assays. They found that ciRNAs such as ci-ankrd52 accumulated in the transcription initiation site of their parental genes and interacted with phospho-Ser5 Pol II-CTD in a sequence-dependent manner to regulate transcription elongation.

In contrast with ciRNAs, EIciRNAs interact with Pol II through a complex mediated by U1 snRNP and regulate transcription initiation [4]. Since no U1 snRNA has been identified in *E. histolytica* [30], the way flicRNAs bind to the Pol II-CTD is unclear. However, U1A interaction with U1-70K and TIA-1 was demonstrated by mass spectrometry of CLIPped mRNPs in this parasite [31]. Considering that TIA-1 is an RBP with a high affinity for U-rich sequences [32] and that the genome of *E. histolytica* is AT-rich genome [33], we searched for linear TIA-1 binding motifs in flicRNA sequences using the bioinformatic tool RBPmap [34]. As expected, we found potential TIA-1 binding sites in all flicRNAs (Appendix A).

Because the U1 snRNP complex directly binds to Pol II through the U1-70K factor [35], it is possible that flicRNAs could interact with the Pol II-CTD via a U1-70k/U1A/TIA-1 complex. However, we did not discard the possibility that flicRNAs interact with Pol II via sequence recognition and RNA tertiary structures that increase their binding capacities to proteins [36].

The binding of U2 snRNP with Pol II-CTD occurs during transcription initiation and elongation [26,27]. Moreover, during elongation, the U6 snRNA base pairs with the 5′ss through its conserved ACAGA box and interacts with U2 snRNA [29,37]. The U6/U2 interaction that leads to the activation of the B complex occurs in synchrony with transcript elongation to 129 nt downstream of the 3′ss [37,38]. Considering these facts, to establish the stage in which flicRNAs would regulate gene expression, we monitored the presence of U2 and U6 snRNAs in the Pol II-CTD CLIPs. The strong signal observed for U2 snRNA, but not for U6 (Figure 1D), allowed us to suggest that flicRNAs regulate transcription initiation and may remain in the complex during elongation before the formation of the B^act^ splicing complex.

Comparative CLIP studies between mutant and wild-type RabX13 introns confirmed that flicRNAs bind to the Pol II complex directly since trophozoites bearing mutant introns showed less interaction between flicX13 and Pol II-CTD (Figure 1C). These findings reassert that the 5′ss GU-rich sequence is implicated in the biogenesis of flicRNAs and takes part in binding motifs formed after circularization that are directly related to the interaction capacities of these molecules. Similar to our findings, chiasmatic-juxtaposed splice sites have been implicated in RBP-circular RNA interactions. Back-splice junction motifs formed in human cardiomyocytes Titin-derived circRNAs are recognized by the auxiliary splicing factor SRSF10, and the mutation of these motifs impaired the cTTN-SRSF10 interaction [39]. The biogenesis of sisRNAS, circRNAs, flicRNAs, and EIciRNAs is tightly linked to the splicing process and machinery, for instance, the auxiliary splicing factors SRSFs (serine/arginine-rich splicing factors); therefore, it is expected that circular RNA biogenesis and modifications impact splicing. When SRSF1, SRSF3, SRSF7, SRSF9, and SRSF10 are recognized and interact with the YTH protein YTHDC1 (also implicated in RNA transport), the alternative splicing patterns are modified in an m6A-dependent manner [40], and splicing patterns can be changed by direct methylation of the 3′ss affecting U2AF35 binding [41]. Furthermore, the intronic sequence m6A methylation is significant [42].

The cytoplasmic localization of flicS14 and flicL12 was unexpected (Figure 1E). However, most intronic circular RNAs have been identified in the cytoplasm of vertebrate cells such as *Xenopus* oocytes. Roughly 80% of these vertebrate sisRNAs show a cytosine residue in the branch point (BP), which could be implicated in the evasion of the Dbr1 processing [19]. Additionally, a GU-rich 5′ss and a C-rich element upstream of the BP are required for ciRNAs to escape the Dbr1 activity [21]. Despite a similar GU-rich 5′ss, flicS14 and flicL12 lariat precursors harbor a different BP configuration, with an adenine instead of a cytosine in the BP site and a lack of C-rich elements upstream of the BP, as expected for the *E. histolytica* genome [33]. Furthermore, C-enriched BP mutants had no significant effect on the biogenesis or localization of flicRNAs [22]. Since nuclear and cytoplasmic *E. histolytica* flicRNAs showed the same cis-elements, additional factors might be implicated in exporting ribosomal flicRNAs to the cytoplasm. The nuclear export of circRNAs leads to a length-dependent evolutionary conserved mechanism that involves a DExH/D box helicase export pathway [43,44]. Moreover, the nuclear export in an N6-methyladenosine (m6A)- dependent manner was reported [45]. The transport of flicRNAs in *E. histolytica* would likely be mediated by similar mechanisms. While m5C methylases exist in *E. histolytica*, there are no reports of m6A methylation in *E. histolytica*. However, several arguments uphold its presence and its possible involvement in flicRNA nuclear export. First, the prevalence and conservation of these RNA modifications and their relationship with RNA processing events also regulates vital cellular mechanisms, even in organisms with low methylation levels [33,43]. Second, m5C and m6A have been linked to replication, transcription regulation, and nuclear export, and both methylation types involved in RNA transport are interrelated. For instance, oxidative stress-induced p21 expression is activated cooperatively by NSUN2-mediated m5C methylation and METTL3/METTL14-mediated m6A methylation [46]. Furthermore, METTL3/METTL14-mediated m6A methylation stimulates NSUN2-mediated m5C methylation, and the m6A reader YTHDF2 can also recognize m5C modification on rRNA [47]. Third, two mechanisms were identified for circRNA nuclear export. In the size-dependent mechanism, whereas UAP56 (DDX39B; EhDEAD18) mediates large circRNA export, URH49 (DDX39A) mediates small circRNAs export [43]. In the RNA element-dependent mechanism, the YTH domain protein recognizes and transports circRNAs bearing m6A-methyl residues in DRACH (D: G/A/U; R: G/A; H: A/C/U) sequence elements [45,48]. In agreement with this, we identified potential m6A methylation DRACH elements on the ribosomal flicS14 and flicL12 (one UAACA element in each flicRNA and an additional UAACT element in flicS14). Furthermore, the m6A methylation machinery interacts with the TREX complex, recruiting m6A readers to DRACH methylation sites [49]. Finally, fourth, UAP56 is part of the TREX complex, and ultimately, they belong to the EJC (Exon Junction Complex). The EJC could be linked to flicRNA export by any of the mechanisms above, provided the machinery copes with flicRNAs whose mean size is 64 nt, and that any of the 15 UAP56 *Entamoeba* orthologs (two of them exist in the oldest Evosea species, *Mastigamoeba balamuthi*) might provide the function of DDX39 in flicRNA recognition; locus EHI_150160 is the best candidate so far.

Although flicRNAs are a small fraction of the circRNA repertoire, and the biogenesis and mechanistic differences among species remain unknown, the presence of intronic circRNAs in multicellular organisms [19] as well as in early-branching unicellular protists [50,51], such as *E. histolytica* [22] and *Euglena gracilis* [52], is the result of their leading roles in early origin and highly conserved regulatory mechanisms in evolution that, as shown in *E. histolytica* flicRNAs, might participate in conserved regulatory networks of gene expression, and probably intron mobility, across species.

Boric acid is a bacteriostatic, fungistatic, and protozoostatic agent reported as a reversible inhibitor of the second transesterification splicing reaction, both in vitro [53,54] and in vivo [22]. Incubation of transfected trophozoites with BA for two hours decreased flicRNAs formation (Figure 2A) and increased the expression of their parental genes (Figure 2B). In addition, our results suggested that BA does not significantly modify the transcription of housekeeping genes (Appendix A). Interestingly, further incubation of trophozoites with BA decreased flicX13 formation even more but did not abolish mRNA levels, suggesting that, at this time point, BA still affects flicRNA production but not de novo mRNA synthesis. Similar outcomes were observed in mutant RabX13 and debranching enzyme transfectants, confirming the post-splicing origin of flicRNAs, as well as their participation in transcription regulation (Figure 2B). The mechanisms underlying these observations have not been fully described. However, boron would form a reversible complex through intermolecular hydrogen bonds with nucleotides of the catalytic splicing site, interfering with the development of the second nucleophilic attack [53] and causing an accumulation of unreleased lariats [22]. Since flicRNAs are bona fide lariat products, this treatment allowed us to conduct a comparative RT-qPCR study to demonstrate that in three different conditions that decrease flicRNAs expression, the level of transcripts of their parental genes is equally affected. As a result, we concluded that flicRNAs constitutively regulate the expression of their parental genes.

We discerned the binding sequences of the flicRNAs-Pol II complex in gene promoters that produce [55] or do not form flicRNAs (Figure 3B,C, respectively). The complexes were bound to the proximal 300 bp upstream of the ATG, similar to the U1/EIciRNA-Pol II complex, which interacts at the promoter sequence 300 bp upstream of the transcription start site [4]. The binding sequences showed multiple core promoter elements previously described in *E. histolytica* [55], such as TATA-like boxes (TATTTAAAG/C), GAAC, and GAAC-like elements in variable locations (Appendix A), which have been associated with the regulation of transcription initiation in this parasite [56,57]. The virulence-associated locus EHI_014170 had an unusual dual promoter spanning 1500 bp (Figure 3A). No defined cis-elements were identified in the −269 to +85 binding site; therefore, the regulation of the flicRNA-Pol II complex would be through an unidentified sequence element in this site, similar to the promoter of the Rab GTPase gene *EhrabB* [58]. However, the second binding site, from −1367 to −913, bore a GAAC box that might constitute an alternative cis-element for the transcriptional control of this locus. The selection of a specific site in this dual promoter would depend on the genomic context, histone modifications, and mutual association or exclusion of transcription factors [59].

The somewhat unexpected association of non-flicRNA-related genes with the flicRNA-Pol II complex (Figure 3C and Figure 4B) suggested that flicRNAs could participate in wider gene regulation networks. This mechanism is widely extended and is controlled at multiple levels. In humans, Zhang et al. observed that ciRNAs associated with different loci. They suggested that they could control expression globally in the cell [21], and Tijsen et al. showed that this control was exerted at the pre-mRNA processing level [39]. In plants, Conn et al. showed that circRNAs controlled transcription and splicing driving floral homeotic phenotypes [60].

Our current view of circular RNA biogenesis, transport, and functions in *E. histolytica* is as follows (Figure 5): Cotranscriptional processing renders mRNA, splicing-derived introns or flicRNAs, that would be circularized by debranching escape [22] and back-splicing-derived circRNAs. Compared to their linear counterparts, these products have longer half-lives and could be m6A and m5C methylated by the METTL3/METTL14/WATP or the TRDNT1 methyltransferase complexes, respectively. In the former complex, a putative METTL3 amoeba ortholog, EHI_013870, was identified by sequence similarity and proteomic analyses [31] (underlined codes denote this fact). Two gene products are annotated as m5C methylases, EHI_098500 and methyltransferase, locus EHI_098730; SRING protein-protein prediction analysis showed that this protein interacts with the EHI_03870 gene product (Appendix A). Next, nuclear flicRNAs (possibly methylated) are recruited at the promoter of different genes by interaction with the B complex spliceosome already placed on the Pol II-CTD, causing up or downregulation of transcriptional initiation. flicRNA-Pol II-CTD interaction could be facilitated by binding the auxiliary splicing factor TIA1 (another member of pre-mRNP [31]) to U-rich flicRNA elements. For methylated linear/circular RNAs, two exporting complexes may be involved, on the one hand, the YTHDC1-SRSF3/NXF/SF3B (EHI_048160), and on the other, the EhDEAD18 (UAP56: EHI_151600)/TREX complex that merges with the EJC complex, becoming fully functional. As stated above, an additional member of the TREX-EJC complex could be EHI_150160. In the cytoplasm, whereas the YTHDF1/3/YTHCD2 complex facilitates the translation of methylated mRNA, methylated mRNA decay is carried out by the YTHDF2 complex, and (possibly methylated) circRNAs would perform their suggested sponging-functions [25]. The function of (possibly methylated) flicRNAs is under investigation.

Notwithstanding the numerous attempts, *E. histolytica* cannot encyst in vitro, impeding understanding of the parasite’s differentiation mechanisms. However, the reptilian parasite *Entamoeba invadens* has been used as a proxy for in vitro encystation studies [61]. Therefore, we searched the abundant transcriptomic data on the *E. invadens* for the orthologs (and paralogs) of the virulence-associated genes analyzed here. We found that most of them are overexpressed either during encystation or excystation (Appendix A). Interestingly, transcripts from loci EIN_391640 and EIN_419420 (orthologs of EHI_169670 and EHI_014170, respectively) also produced circular RNA molecules [25], suggesting a possible role of such molecules both in virulence and trophozoite to cyst differentiation.

In conclusion, we found that flicRNAs showed heterogeneous cellular localization. While the function of cytoplasmic flicRNAs remains uncovered, nuclear flicRNAs regulate the transcription initiation of genes in a widespread fashion through a complex with Pol II on the promoter sequences in *E. histolytica*, impacting the parasite’s virulence, encystment, and excysting.

## 4. Materials and Methods

**Cell cultures and drug treatments.***E. histolytica* trophozoites (HM1: IMSS strain) were axenically grown at 37 °C in Trypticase-yeast extract- iron serum (TYI-S-33) medium and harvested as described [62]. For the inhibition of the second step of splicing assays, 10^6^ log phase (48 h) wild-type or transfectant trophozoites per experimental point were treated with different concentrations of BA (pH 7.9) [53] during 1.5 h. After treatments, the cells were harvested, and the total RNA or DNA was isolated and analyzed by reverse transcription (RT), followed by quantitative or endpoint polymerase chain reaction (PCR), respectively.

**RNA and DNA isolation.** The genomic DNA (gDNA) and total RNA were isolated using the TRIzol Reagent specified by the manufacturer (Invitrogen). For gDNA purification, the preparations were treated with RNases T1 and A. When specified, for total or circular RNA isolation, the reactions were treated with RQ1 DNase (Promega) or RNase R (Epicentre), respectively.

**Retrotranscription and polymerase chain reactions.** All detailed PCR conditions and primers’ sequences are listed in Appendix A. RabX13 (EHI_065790), U6 snRNA (U43841), U2 snRNA (BK006130), HSP70 (EHI_052860), and EhActin (EHI_107290) gene expression was monitored by RT-PCR using specific primer pairs, M-MLV retro transcriptase and Taq DNA Polymerase, as specified by the manufacturer (Invitrogen). Specific outward-facing primer pairs targeted to the introns of the *E. histolytica* genes RabX13, RpL12 (EHI_191750), RpS14 (EHI_074090), ClcB (EHI_186860), and the virulence-associated loci EHI_169670, EHI_014170, and EHI_192510, were designed as described [24] for the detection of circular RNA molecules. The primers were used in circular RT-PCR using the conditions above. Retro-transcription reactions were carried out using 5 µM of actinomycin D (SigmaAldrich, Toluca, México). The drug was added immediately after the denaturing step [63]. PCR products were resolved in 2.5% agarose gels. The Kappa Sybr Fast Universal One-Step RT-qPCR kit (Sigma Aldrich) was used for quantitative RT-PCR with 10 ng of cDNA input in 10 µL. EhRNA Pol II (EHI_056690) was used as a normalizer to calculate all the transfectants’ relative expression based on the Livak method [64].

**Plasmid constructs and Entamoeba transformants.** Using *E. histolytica* gDNA as a template, and the respective primer pairs in which the SmaI and XhoI restriction sites were included, the full-length (HA-Pol II) and C-terminus (HA-Pol II-CTD) of the major subunit of RNA polymerase II were amplified by PCR and ligated into the SmaI/XhoI-digested pEhExHA (HA-V) expression plasmid able to express N-terminal hemagglutinin (HA)-tagged fusion protein [65]. All PCR products were also cloned into the pCR2.1 plasmid vector (Invitrogen) and sequenced. Moreover, transfectants of HA-tagged wild-type and mutant RabX13 minigenes, as well as the C-terminus deleted Dbr1 (Dbr1ΔC) [22], were established. Plasmids were transfected into *E. histolytica* HM1: IMSS trophozoites as described [66]. To avoid lethality, 24 h after transfection, the expression of HA-Dbr1ΔC fusion proteins in transient transfectants was induced by the addition of 1.5 µg/mL G418 for 48 h, and HA-Pol II was stably maintained at 4 µg/mL G418. As above, HA-Pol II-CTD transfectants were established by culturing in 3, 6, or 10 µg/mL G418. The presence of the tagged genes in the transfectant strains was monitored by PCR with genomic DNA using Pol II-CTD-, Pol II- and ΔGU-specific primers.

**Western blots.** HA-Pol II and actin were detected by Western blot using the anti-HA (Covance) and anti-actin primary antibodies, diluted 1:10,000 and 1:3000, respectively, from nuclear extracts of *E. histolytica* trophozoites. The protein concentration was measured by the Bradford method. Using standard techniques, the proteins (25 μg per lane) were separated by 12% SDS-PAGE and electro-transferred to a 0.2 μm nitrocellulose membrane (BIO-RAD). Primary antibodies were detected using goat anti-mouse IgG peroxidase conjugate (Invitrogen).

**Immunofluorescence and laser confocal microscopy.** Protein lysates and immunolocalizations by confocal microscopy were carried out, essentially, as described [67]. For nuclear staining, 4′,6-diamino-2-phenylindole (DAPI) was included in the mounting medium.

**CLIP assays.** Amoeba transformants (pHA, pHA-Pol II-CTD, pHA-ΔGU, and pHA-Pol II-CTD/ΔGU) were irradiated with UV light in a Stratalinker^®^ UV Crosslinker 2400 for 30 min in PBS 1X. Then, starting from nuclear extracts, immunoprecipitations with anti-HA agarose were carried out as described in [31,67]. Aliquots of the nuclear extracts were saved as input. RNA was purified from eluates, and, to rule out artifacts and confirm flicRNAs’ identity, the RNA samples were treated with RNase R before the divergent RT-PCR (Appendix A shows representative examples). No significant differences between the amplification of flicRNAs in treated and untreated conditions were observed, indicating that they were bona fide flicRNAs.

**Chromatin immunoprecipitation.** ChIP assays were carried out as previously described [68] with modifications. HA-Pol II and PHA transfected trophozoites (2 × 10^6^) were harvested and incubated in 1% formaldehyde for 10 min at room temperature. Cells were pelleted at 1500 rpm for 15 min and washed three times with cool PBS 1X, Buffer I (HEPES 10 mM, pH 6.5; EDTA 10 mM; EGTA 0.5 mM; Triton X-100 0.25%) and Buffer II (HEPES 10 mM, pH 6.5; EDTA 1 mM; EGTA 0.5 mM; NaCl 200 mM). The cells were resuspended in lysis buffer with a protease inhibitors cocktail (Roche) and sonicated to obtain 500 bp DNA fragments. The extract was clarified by centrifugation and incubated for 30 min in chromatin solubilization buffer (Tris 20 mM pH 8.1, NaCl 150 mM, EDTA 2 mM, Triton X-100 1%, salmon sperm DNA 50 μg/mL, tRNA 100 mg/mL), bovine serum albumin 1 mg/mL, and 50 μL rec-Protein G-Sepharose (Invitrogen). A 100 µL sample of the supernatant corresponding to soluble chromatin (0.5 mL) was saved as input. An aliquot of 500 µL was incubated with anti-HA antibodies (2 μg) overnight at 4 °C and then with rec-Protein G-Sepharose for 2 h, at 4 °C. The beads were sequentially washed with buffer III (Tris 20 mM pH 8.1, EDTA 2 mM, Triton X-100 1%, SDS 0.1%, NaCl 150 mM), buffer IV (Tris 220 mM pH 8.1, EDTA 2 mM, Triton X-100 1%, SDS 0.1%, NaCl 500 mM), LiCl buffer (Tris 10 mM pH 8.1, LiCl 250 mM, Nonidet P-40 1%, deoxycholate 1%, EDTA 1 mM), and buffer TE (Tris 10 mM pH 8.1 and EDTA 1 mM). Immune complexes were retrieved by incubation in elution buffer (SDS 0.1%, NaHCO_3_ 100 mM, 20 μg/mL glycogen) for 15 min at room temperature. Reverse cross-linking was conducted from the TE and elution fractions. Eluates were incubated with 0.2 M NaCl for 4 h at 65 °C and divided into two aliquots. The first aliquot was treated with proteinase K and RNase A for promoter DNA identification, and the second aliquot was treated with proteinase K and RNase R for flicRNAs identification. A control sample was treated with DNase RQ1. DNA was extracted, and semi-quantitative RT-PCR analysis of the selected promoters was performed using 2 μL per reaction and specific oligonucleotides (Appendix A).

**Statistical analyses.** The values reported in the graphs represent averages of at least two independent experiments, significance being (*) *p* < 0.001. The intensity of gel bands was measured using NIH ImageJ software, and the RT-qPCR data were analyzed using the 2^−ΔΔCT^ method [64]. GraphPad Prism version v5.01 for Windows, GraphPad Software, San Diego, California USA was used to analyze results, using the Student’s t-test and One-Way ANOVA. The mean value of each variant in the empty vector controls, the absence of boric acid, or at the zero time points was set as 1.

## Figures and Tables

**Figure 1 ncrna-08-00012-f001:**
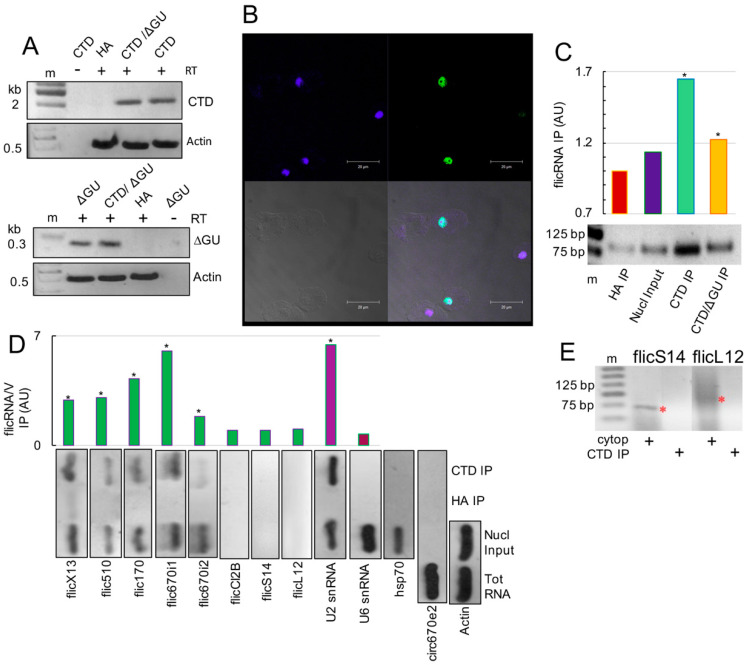
flicRNAs associated with RNA Pol II-CTD domain. (**A**) PCR amplifications were used to verify the establishment of single and double amoeba transfectants. Appropriate primers were used to amplify the 2240 bp HA-Pol II-CTD-specific (CTD) and 287 bp HA-ΔGU-specific fragments in single and HA-Pol II-CTD/ΔGU (CTD/ΔGU) double transfectants. DNA fragments were resolved in 1 or 2% agarose gels. HA-V transfectants were used as negative controls, and actin was amplified as a loading control. Minus RT controls were also carried out. M, molecular weight marker. (**B**) Two nuclear localization signals (KLKSENKLEIRRKNGIK and LRKKNFKSIEERLSSKQGRL, residues 178–194 and 314–333, respectively) import the Pol II-CTD domain into the nucleus of amoeba transformants. The merged immunofluorescence confocal microscopy (bottom right panel; 100× magnification) representative image shows signals for anti-HA antibodies and FIT-C conjugated secondary antibodies (green channel, top right panel; 60% positive cells of 3 fields, 100 cells each); DAPI-stained nuclear DNA (blue channel, top left panel); phase contrast (bottom left panel); scale bar = 20 μm; 75% at least three areas of 100. (**C**) CLIP validation: densitometric quantitation of flicX13 amplified using circular RT-PCR, from CLIP RNA samples of HA IP, nuclear input, Pol II-CTD IP (CTD IP), and Pol II-CTD/ΔGU (CTD/ΔGU IP) immunoprecipitations. Amplified products were resolved in 2.5% agarose gels (bottom). (**D**) Different flicRNAs were amplified as in (**C**). To monitor the cotranscriptional nature of CLIPs, the spliceosomal U2 snRNA (U2) was amplified, and U6 snRNA (U6) was used as the corresponding negative control. The Hsp70 transcript was amplified to test the specificity of the HA antibodies. The cytoplasmic circular RNA circ670e2 was amplified to monitor cellular fractionation, and actin was amplified as the circ670e2 loading control. The plot shows the average abundance of amplicons of two independent experiments. (**E**) Divergent RT-PCR was used to amplify ribosomal flicRNAs RpS14 and RpL12 from cytosolic RNA fractions (red asterisks) and RNA purified from Pol II-CTD CLIPs (CTD IP). Products were resolved in 2.4% agarose gels. m, molecular weight marker. Statistically significant results (ANOVA, Turkey test *a posteriori*. *p* ≤ 0.01, *).

**Figure 2 ncrna-08-00012-f002:**
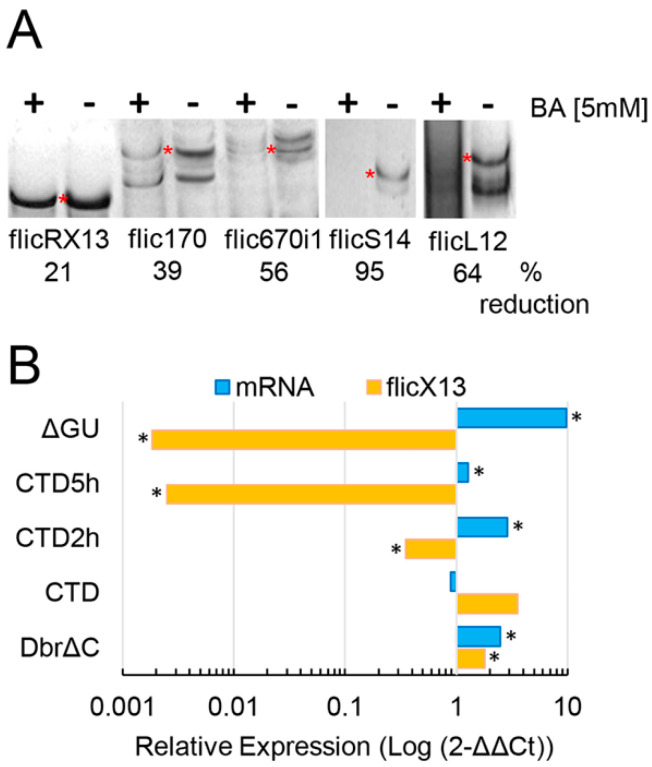
Inverse correlation of flicX13 and RabX13 gene expression. (**A**) 5 mM boric acid (BA) reduces the production of flicRNAs flicX13 (lanes 1–2), flic170 (lanes 3–4), and flic670i1 (asterisks in lanes 5–6), assessed by circular RT-PCR using specific primers and total RNA isolated from BA treated (+) and untreated (−) amoebas. DNA fragments were visualized on 2.5% agarose gels. Densitometric analyses of the results are shown above. (**B**) RT-qPCR was used to monitor the effect of BA, for 2 and 5 h, on the production of flicX13 and the expression of mRabX13 in Pol II-CTD transfectants (CTD). As controls, we used untreated DbrΔC and ΔGU trophozoites. They elicit flicRNA accumulation/slight mRNA reduction and flicRNA reduction/mRNA accumulation, respectively. Relative expression of RNA variants is presented compared to one in the HA control. Logarithmic values are the average of three independent experiments; statistically significant results (ANOVA and Turkey test *a posteriori. p* ≤ 0.01, *) are shown.

**Figure 3 ncrna-08-00012-f003:**
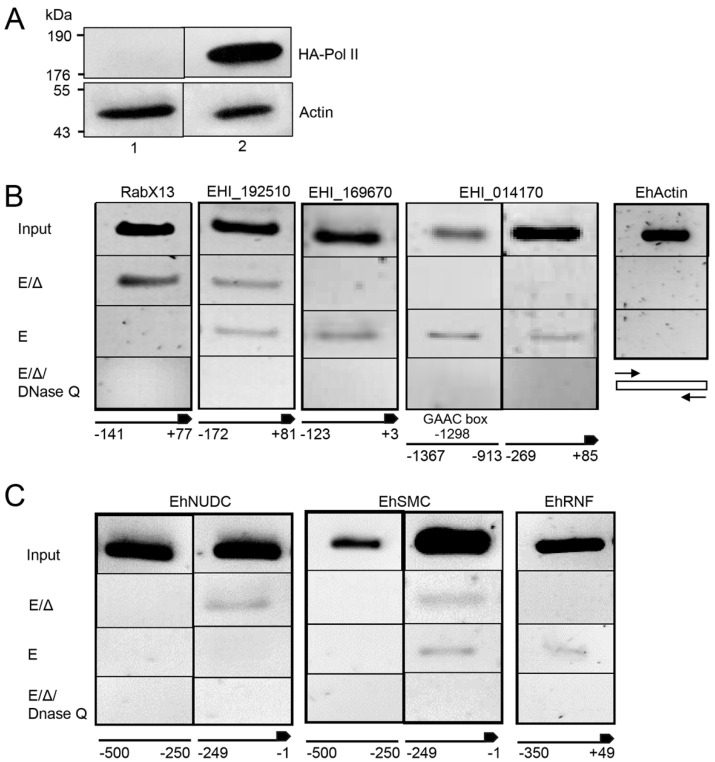
RNA Pol II binds to gene promoters regardless of their flicRNA-forming capacity. (**A**) *E. histolytica* trophozoites were transiently transfected with the empty vector (pHA, lanes 1) or with the HA-tagged major subunit of the RNA polymerase II (HA-Pol II, lanes 2) and selected with low concentrations of G418 (4 µg/mL). Overexpression of HA-Pol II was monitored by Western blot using anti-HA antibodies compared to anti-actin (EHI_107290). Using these amoeba transfectants, ChIP assays detect RNA Pol II binding sites on selected flicRNA-forming promoters (**B**), as well as on promoters that do not form flicRNAs (**C**). PCR amplified DNA fragments were resolved in 2.5% agarose gels. Input fractions of sonicated DNAs are shown. Tris/SDS elution (E), heat reversed-crosslinked (ΔE), and DNase Q treated (ΔEQ) samples were analyzed. In each step of the ChIP assay, the actin coding region was used as a control. The position of the primers relative to their promoters and the first codon are shown. Based on their sequence motifs in large promoters, two DNA fragments were analyzed.

**Figure 4 ncrna-08-00012-f004:**
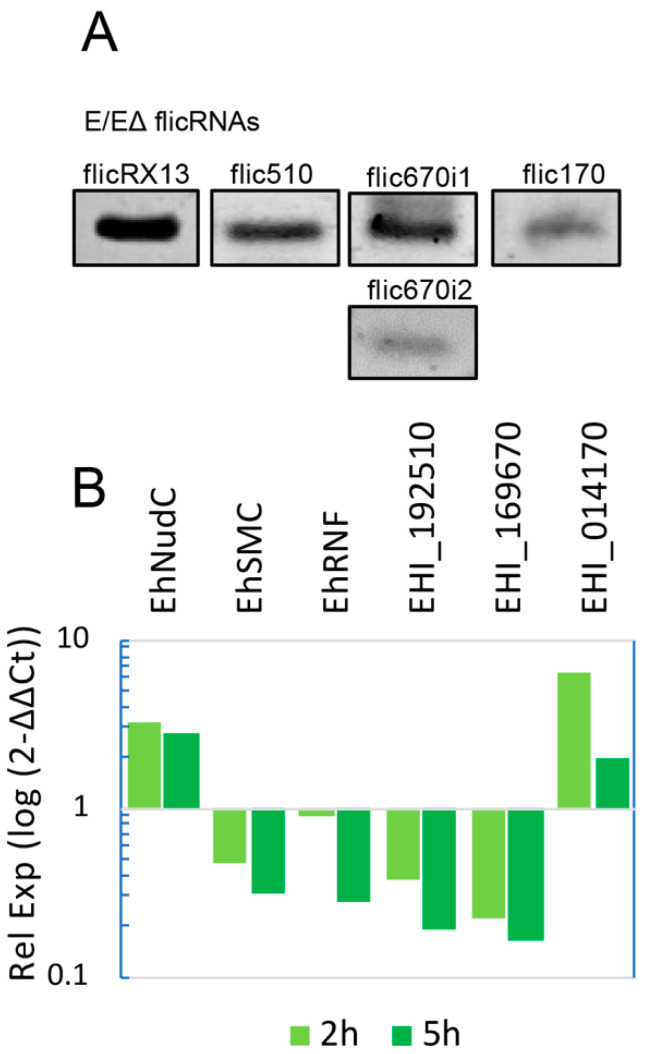
flicRNAs are found on the promoters of their parental genes and alter the expression of flicRNA-forming and no flicRNA-forming genes. (**A**) RNA was extracted from ΔE fractions of HA-Pol II ChIP aliquots and used to perform circular RT-PCR to detect immunoprecipitated flicRNAs. (**B**) After 2 h or 5 h boric acid-induced flicRNA production reduction, the expression of the non-flicRNA-forming genes, EhNudC, EhSMC, and EhRNF (2 h sample only), and the flicRNA-forming genes, EHI_192510, EHI_169670, and EHI_014170, was analyzed by RT-qPCR, compared to untreated control amoebas, set as 1 in each case. The logarithmic plot shows that all expression profiles are statistically significant (ANOVA and Turkey test a posteriori. *p* ≤ 0.01).

**Figure 5 ncrna-08-00012-f005:**
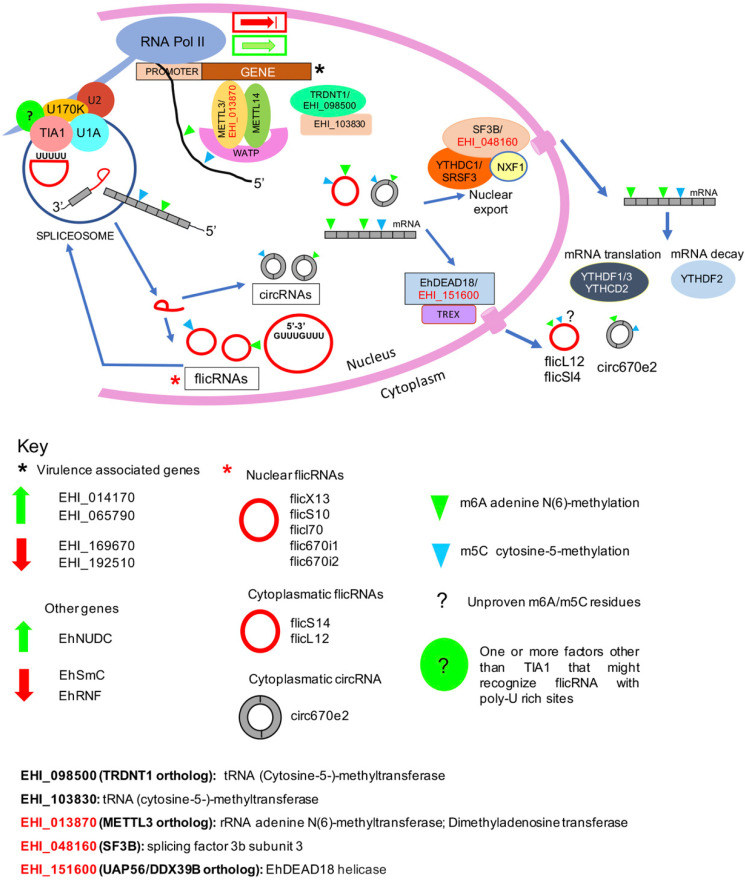
*Entamoeba histolytica* mRNA and circular RNA biogenesis, transport, and functions. The model described in the main text depicts our current understanding of the protein complexes involved in each process. Components of the spliceosome previously described in our group [31]. Putative methylases and nuclear export proteins that have been previously identified by proteomic analysis of pre-messenger ribonucleoparticles are highlighted in red. Activation/silencing (green upwards/red downwards arrows, respectively) of virulence-related and nuclear gene transcripts (black asterisk) are indicated, as well as the flicRNAs (red asterisk) described here. Green and blue arrowheads indicate adenine N6-methylation and cytosine-5-methylation, respectively. The question mark denotes unproven methylation events. A green bubble with a question mark represents additional factor(s) facilitating flicRNA-spliceosome-Pol II-CTD interaction involved in transcriptional regulation.

## Data Availability

Not applicable.

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
