# Peer review of "Promoter-Bound Full-Length Intronic Circular RNAs-RNA Polymerase II Complexes Regulate Gene Expression in the Human Parasite Entamoeba histolytica"

_ncrna, 2022, doi:10.3390/ncrna8010012_

Round 1

Reviewer 1 Report

This manuscript by García-Lerena et. al. examines the role of flicRNAs in gene expression regulation in coordination with RNA pol II complex in E. histolytica. Overall, experimental plans and results concur with the potential gene regulatory functions of flicRNAs and this work can provide insights into important aspects of E. histolytica post-transcriptional gene regulation. However, there are some important revisions required before this manuscript is accepted for publications. One major problem in the entire manuscript is that authors do not elaborately discuss the reasons they chose to do a particular experiment, why the reagents they used were vital for this purpose and how the results support their hypothesis. Some particular issues and pointers are listed here:

  1. Refer ‘CTD’ as PolII-CTD to make it consistent and relevant throughout the manuscript
  2. In Results 2.1- ‘CLIP assays reveal flicRNAs-RNA polymerase II interactions’, clearly organize the sentences to justify the reasons why each of these constructs and its relevance to each of the experiments done. In current version, all ideas/thoughts are jumbled up together making it difficult to understand which experiment was done for what kind of reason? For example: “we first guaranteed the exogenous expression ….. CLIP and RT-qPCR comparative assays respectively”. What was the reason to make the CDT-HA and full-length PolII constructs? Are these experiments related to where pExEhHA and the RabX13 minigene construct was used? Why creating HA-CTD/∆GU double transfectants and HA-Dbr1∆C were necessary?
  3. Microscopy images in Figure 1B are difficult to interpret. Authors should provide more legible zoomed in version of images with a replicate count of how many times they observed these foci in repeat experiments.
  4. If there’s no significant change in expression profiles, in figure 2, why include that part?
  5. Not consistent with spelling “amoeba”. For example, Figure 1A spells it “ameba” and Figure 1B spells it “amoeba”.
  6. Were there any proteins in the mass spec data that could support the m6A-dependent nuclear export of flicRNA as stated in the discussion this is a likely occurrence? Could m6A influence splicing pattern?

7. Figure 4 is claimed, in the discussion, to show a global transcription alteration but a select few genes were chosen. It doesn’t appear to give a “global” analysis. 

Author Response

Reply to Reviewer 1.

Our sincere thanks to the Reviewers for their helpful comments.

Issues addressed.

  1. Refer ‘CTD’ as PolII-CTD to make it consistent and relevant throughout the manuscript

A: This issue has been corrected throughout the text. In addition, in the figure legends, amends were introduced to make consistent the changes with the figures (to avoid crowding the figures were not modified; we are sorry for this).

  1. In Results 2.1- ‘CLIP assays reveal flicRNAs-RNA polymerase II interactions’, clearly organize the sentences to justify the reasons why each of these constructs and its relevance to each of the experiments done. In current version, all ideas/thoughts are jumbled up together making it difficult to understand which experiment was done for what kind of reason? For example: “we first guaranteed the exogenous expression ….. CLIP and RT-qPCR comparative assays respectively”. What was the reason to make the CDT-HA and full-length PolII constructs? Are these experiments related to where pExEhHA and the RabX13 minigene construct was used? Why creating HA-CTD/∆GU double transfectants and HA-Dbr1∆C were necessary?

A: We clarified the need to use each one of the constructs. Next, we organized the description of the results to clarify the arguments and line of thoughts, lines 92-166, as follows:

CLIP and ChIP assays were used to assess flicRNAs-Pol II association in solution and on the promoters of different genes. To this end, we first established hemagglutinin (HA)- tagged amoeba transformants of the Pol II-CTD (to monitor flicRNA- Pol II-CTD association), as well as the full-length major subunit of Pol II (to observe promoter-bound flicRNA-Pol II associations). Assays were compared to the empty vector (pExEhHA) Entamoebatransformant controls or to HA-tagged amoeba transformants with decreased or increased RabX13 flicRNA expression: respectively, ∆GU (RabX13 minigene 5´ss intronic mutant), and Dbr1∆C (intron lariat debranching-deficient enzyme) [22]. To control efficient CLIP signal detection, the expression of the tagged genes in the single and double-transfectant strains was first verified by RT- PCR using primers to detect HA- Pol II-CTD and HA-∆GU DNA fusions (Figure 1A).

Using immunofluorescence and confocal microscopy assays, we next showed that the expressed Pol II-CTD conserved its nuclear localization in transfected trophozoites using FIT-C conjugated secondary antibodies. Whereas no signal was detected in the cytoplasm, significant punctate signals were found in the nucleus of 60 % of the cells as expected from two nuclear localization signals included in the Pol II-CTD domain insert (Figure 1B).

To validate our CLIP methods and further analyze the implication of the flicRNAs in transcription regulation, the model flicRNA, flicX13, was monitored using outward-facing primers in circular RT-PCR [24] after CLIP assays were conducted in Pol II-CTD/∆GU trophozoites that exogenously express both Pol II-CTD and the RabX13 minigene bearing a mutation in the 5’ss of the intron which decreases flicRNA biogenesis [22]. As expected, compared to the empty vector and nuclear input, we observed increased recovery of flicX13 in Pol II-CTD immunoprecipitations, which correspondingly decreased in the Pol II-CTD/∆GU double transfectants (Figure 1C). These results validate our approach and indicate that flicRNA-mediated transcriptional regulation might be exerted via flicRNA- Pol II-CTD interaction.

CLIP assays were then carried out to detect flicRNAs of the RabX13 gene, the virulence-associated loci EHI_169670 (both introns), EHI_014170, EHI_192510, and Clc2B, and of the ribosomal proteins RpS14 and RpL12: flicX13, flic670i1, flic670i2, flic170, flic510, flicCl2B, flicS14 and flicL12, respectively. CLIP specificity was confirmed by the lack of amplification of the mRNA of a non-related gene (Hsp70) as well as the exonic circRNA circ670e2 [25] (exonic circRNAs are mainly cytoplasmic [6]) (Figure 1D). Actin was amplified to normalize circ670e2 amplification between fractions.

 We found that flicX13, flic670i1, flic670i2, flic170, and flic510 were bound to the Pol II-CTD, suggesting their participation in regulating RNA Pol II in cis (Figure 1D). To gain insights into this possibility, we reasoned that since U2 snRNA interactions with phospho-Serine 5 (P-Ser5) residues in the Pol II-CTD [26, 27] are required for cotranscriptional spliceosome assembly on the nascent pre-mRNA as well as interactions of U6 snRNA with U2 snRNA and the 5’ss to activate the splicing complex B [28, 29] the search for U2 and U6 snRNAs as part of the Pol II cotranscriptional complex was required. A strong signal of U2 snRNA but not of U6 snRNA (Figure 1D) was observed, suggesting that flicRNA- Pol II-CTD interaction occurs during transcription initiation/elongation before complex B activation.

Since no CLIP signal was detected for flicCl2B, flicS14, and flicL12, we did identify flicS14 and flicL2 in total RNA from trophozoites (Fig. 2A), their absence in the Pol II-CTD immunoprecipitants and the nuclear input suggested a cytoplasmatic localization of these ribosomal flicRNAs. To determine their subcellular localization, purified cytosolic RNA was used to perform divergent RT-PCR assays. Both flicRNAs were detected in the cytoplasmatic fraction (Figure 1E). flicS14 appeared in a band at 73 bp while flicL12 showed an unspecific amplification around 100 bp.

            In addition, in Results 2.2 the use of the constructs was explained also, lines 177-180:

The decrease of flicX13 expression is comparable to the flicRNA biogenesis-deficient ∆GU transfectants.Conversely, the increase of flicX13 expression is similar to that of Dbr1∆C amoeba transfectants which elicit flicRNA accumulation [22].

  1. Microscopy images in Figure 1B are difficult to interpret. Authors should provide more legible zoomed in version of images with a replicate count of how many times they observed these foci in repeat experiments.

A: We substituted the figure with the merged field only and the number of cells with signals was included in the main text:

Whereas no signal was detected in the cytoplasm, significant punctate signals were found in the nucleus of 60 % of the cells as expected from two nuclear localization signals included in the Pol II-CTD domain insert (Figure 1B).

and in the figure legend, lines 100-105:

(B) Two nuclear localization signals (KLKSENKLEIRRKNGIK and LRKKNFKSIEERLSSKQGRL, residues 178-194 and 314-333, respectively) import the Pol II-CTD domain into the nucleus of amoeba transformants. The merged immunofluorescence confocal microscopy (100X magnification) representative image shows signals for anti-HA antibodies and FIT-C conjugated secondary antibodies (green channel; 60 % positive cells of 3 fields, 100 cells each); DAPI-stained nuclear DNA (blue channel); phase contrast; scale bar = 20 mm; 75 %at least three areas of 100.

  1. If there’s no significant change in expression profiles, in figure 2, why include that part?

A: This part was described in the text only, and the figure was sent to Supplementary data, lines 181-183.

To discard that BA treatments could affect global transcription, we monitored the expression of constitutive transcripts. Expression of Actin and Hsp70 did not change significantly in H3BO3 treated amoebas compared to untreated cells (Figure S1).

  1. Not consistent with spelling “amoeba”. For example, Figure 1A spells it “ameba” and Figure 1B spells it “amoeba”.

A: this issue was corrected, “amoeba” was used throughout.

  1. Were there any proteins in the mass spec data that could support the m6A-dependent nuclear export of flicRNA as stated in the discussion this is a likely occurrence? Could m6A influence the splicing pattern?

A: In Discussion, two sections were included to address these concerns, in lines 295-306:

Back-splice junction motifs formed in human cardiomyocytes Titin-derived circRNAs are recognized by the auxiliary splicing factor SRSF10, and mutation of these motifs impaired the cTTN-SRSF10 interaction [39]. The biogenesis of sisRNAS, circRNAs, flicRNAs, and EIciRNAs is tightly linked to the splicing process and machinery, for instance, the auxiliary splicing factors SRSFs (Serine/Arginine-rich splicing factors), therefore it is expected that circular RNA biogenesis and modifications impact splicing. When SRSF1, SRSF3, SRSF7, SRSF9, and SRSF10 are recognized and interact with the YTH protein YTHDC1 (also implicated in RNA transport), the alternative splicing patterns are modified in an m6A-dependent manner [40], and splicing patterns can be modified by direct methylation of the 3’ss affecting U2AF35 binding [41]. Furthermore, intronic sequences m6A methylation is significant [42].

and in lines 322-349:

The transport of flicRNAs in E. histolytica would likely be mediated by similar mechanisms. While m5C methylases exist in E. histolytica, there are no reports of m6A methylation in E. histolytica. However, several arguments uphold its presence and its possible involvement in flicRNA nuclear export. First, the prevalence and conservation of these RNA modifications and their relationship with RNA processing events also regulate vital cellular mechanisms, even in organisms with low methylation levels [33, 43]. Second, m5C and m6A have been linked to replication, transcription regulation, and nuclear export, and both methylation types involved in RNA transport are interrelated. For instance, oxidative stress-induced p21 expression is activated cooperatively by NSUN2-mediated m5C methylation and METTL3/METTL14-mediated m6A methylation [46]. Also, METTL3/METTL14-mediated m6A methylation stimulates NSUN2-mediated m5C methylation, and the m6A reader YTHDF2 can also recognize m5C modification on rRNA [47]. Third, two mechanisms have been identified for circRNA nuclear export. In the size-dependent mechanism, whereas UAP56 (DDX39B; EhDEAD18) mediates large circRNA export, URH49 (DDX39A) mediates small circRNAs export [43] Wan et al., 2018). In the RNA element-dependent mechanism, the YTH domain protein recognizes and transports circRNAs bearing m6A-methyl residues in DRACH (D: G/A/U; R: G/A; H: A/C/U) sequence elements [45, 48]. In agreement with this, we have identified potential m6A methylation DRACH elements on the ribosomal flicS14 y flicL12 (one UAACA element in each flicRNA and an additional UAACT element in flicS14). Furthermore, the m6A methylation machinery interacts with the TREX complex, recruiting m6A readers to DRACH methylation sites [49]. And fourth, UAP56 is part of the TREX complex, and ultimately, they belong to the EJC (Exon Junction Complex). The EJC could be linked to flicRNA export by any of the mechanisms above, provided the machinery copes with flicRNAs whose mean size is 64 nt, and that any of the 15 UAP56 Entamoeba orthologs (two of them in the oldest Evosea species, Matigamoeba balamuthi) might provide the function of DDX39 in flicRNA recognition; locus EHI_150160 is the best candidate so far.

  1. Figure 4 is claimed, in the discussion, to show a global transcription alteration but a select few genes were chosen. It doesn’t appear to give a “global” analysis. 

A: Not only in Figure 4, but in all sections (lines 28, 88, 227, and 454) the word global was substituted for general.

Reviewer 2 Report

The manuscript by García-Lerena and coworkers describes an interesting regulatory mechanism based on the interaction of RNApol II with circRNAs in the promoter region. The results were obtained by using Entamoeba sp as a model. The contents are very relevant and interesting for the journal readers.

The manuscript is technically sound, and the experimental procedures are well described. Conclusions are well supported by data.

I have only minor comments for the improvement of the contents before its publication.

Point to point comments

1.- Please complete the acknowledgements and conflicts of interest section of the manuscript. The current version contains the default text available from the publisher.

2.- Species names should be written in italics. Please check along the whole text for non-italicized text referring to specific names.

3.- Include an additional figure that could be also as “graphical abstract”, summarizing the findings of the manuscript, and describing the molecular mechanism proposed for the regulatory effect of the flicRNA in Entamoeba.

4.- Expand the discussion including information regarding the presence of this mechanism of transcriptional regulation mediated by RNApol-circRNA interaction in other organisms different from parasites. How extended is this mechanism? Are there some examples in higher eukaryotes?

Author Response

Reply to Reviewer 2

We are grateful to the Reviewers for their comments.

Point by point answers

1.- Please complete the acknowledgements and conflicts of interest section of the manuscript. The current version contains the default text available from the publisher.

A: The acknowledgment section was eliminated since the Funding section was completed. The conflict-of-interest section was completed.

2.- Species names should be written in italics. Please check along with the whole text for non-italicized text referring to specific names.

A: The scientific names of all the species mentioned have been corrected and italicized in the text. See for example lines 121, 310, and 347.

3.- Include an additional figure that could be also a “graphical abstract”, summarizing the findings of the manuscript, and describing the molecular mechanism proposed for the regulatory effect of the flicRNA in Entamoeba.

A: Figure 5 was included to summarize our current understanding of the regulatory mechanisms, effects, and functions of flicRNAs and circRNAs in Entamoeba. See lines 403-439.

4.- Expand the discussion including information regarding the presence of this mechanism of transcriptional regulation mediated by RNApol-circRNA interaction in other organisms different from parasites. How extended is this mechanism? Are there some examples in higher eukaryotes?

A: The discussion section was expanded to include information on RNA Pol II-circRNA interactions involved in transcription regulation in humans and plants. See lines 396-401.